# Fluorescence-Guided Surgical Techniques in Adult Diffuse Low-Grade Gliomas: State-of-the-Art and Emerging Techniques: A Systematic Review

**DOI:** 10.3390/cancers16152698

**Published:** 2024-07-29

**Authors:** Thiebaud Picart, Arthur Gautheron, Charly Caredda, Cédric Ray, Laurent Mahieu-Williame, Bruno Montcel, Jacques Guyotat

**Affiliations:** 1Department of Neurosurgery, Hôpital Neurologique Pierre Wertheimer, Groupe Hospitalier Est, Hospices Civils de Lyon, 59 Boulevard Pinel, 69500 Bron, France; 2Faculty of Medicine Lyon Est, Université Claude Bernard Lyon 1, 8 Avenue Rockefeller, 69003 Lyon, France; 3Cancer Research Centre of Lyon (CRCL) Inserm 1052, CNRS 5286, 28 Rue Laennec, 69008 Lyon, France; 4Laboratoire Hubert Curien UMR 5516, Institut d’Optique Graduate School, CNRS, Université Jean Monnet Saint-Etienne, 42023 Saint-Etienne, France; arthur.gautheron@creatis.insa-lyon.fr; 5CREATIS CNRS, Inserm, UMR 5220, U1294, INSA-Lyon, Université Claude Bernard Lyon 1, UJM-Saint Etienne, 69100 Lyon, France; charly.caredda@creatis.insa-lyon.fr (C.C.); cedric.ray@univ-lyon1.fr (C.R.); mahieu@creatis.insa-lyon.fr (L.M.-W.); bruno.montcel@univ-lyon1.fr (B.M.)

**Keywords:** 5-ALA, biopsy, confocal laser endomicroscopy, fluorescein sodium, fluorescence-guided resection, diffuse low-grade gliomas, spectroscopy

## Abstract

**Simple Summary:**

In the era of targeted therapies, achieving a maximal safe resection still remains a critical prognosis factor in diffuse low-grade gliomas and represents the goal to reach for neurosurgeons who manage these tumors. Whereas fluorescence-guided surgery, using 5-aminolevulinic acid or even fluorescein sodium significantly increases the extent of resection of high-grade gliomas, its relevance for the resection of low-grade gliomas is very limited. However, new intraoperative techniques such as spectroscopic detection of 5-aminolevulinic acid-induced fluorescence or confocal laser endomicroscopy have been developed in order to increase the sensitivity of fluorescence detection in low-grade gliomas or to generate optic biopsies, respectively. Therefore, the goal of this review was to sum up the limitations of macroscopic fluorescence detection in low-grade gliomas and to highlight how new technologies likely to be implemented in the surgical routine in the near future, could overcome these limitations.

**Abstract:**

Diffuse low-grade gliomas are infiltrative tumors whose margins are not distinguishable from the adjacent healthy brain parenchyma. The aim was to precisely examine the results provided by the intraoperative use of macroscopic fluorescence in diffuse low-grade gliomas and to describe the new fluorescence-based techniques capable of guiding the resection of low-grade gliomas. Only about 20% and 50% of low-grade gliomas are macroscopically fluorescent after 5-amino-levulinic acid (5-ALA) or fluorescein sodium intake, respectively. However, 5-ALA is helpful for detecting anaplastic foci, and thus choosing the best biopsy targets in diffuse gliomas. Spectroscopic detection of 5-ALA-induced fluorescence can detect very low and non-macroscopically visible concentrations of protoporphyrin IX, a 5-ALA metabolite, and, consequently, has excellent performances for the detection of low-grade gliomas. Moreover, these tumors have a specific spectroscopic signature with two fluorescence emission peaks, which is useful for distinguishing them not only from healthy brain but also from high-grade gliomas. Confocal laser endomicroscopy can generate intraoperative optic biopsies, but its sensitivity remains limited. In the future, the coupled measurement of autofluorescence and induced fluorescence, and the introduction of fluorescence detection technologies providing a wider field of view could result in the development of operator-friendly tools implementable in the operative routine.

## 1. Introduction

In essence, diffuse gliomas, independently of the grade, are infiltrative tumors which spread far beyond an MRI anomaly and whose margins are not distinguishable from the non-tumoral brain parenchyma by the naked eye, even with the help of white-light microscopy [1,2]. Intraoperative fluorescence is one of the surgical tools that have been developed during the past decades in order to facilitate tumor delineation and maximize the extent of high-grade glioma (HGG) resection [3,4,5]. Fluorescence-guided surgery is gaining more and more attention worldwide [6,7,8,9], notably since the approval of 5-Amino-Levulinic Acid (5-ALA) by the American Food and Drug Administration in 2017 [10]. The performances of 5-ALA fluorescence-guided surgery for the resection of glioblastomas have been studied in two randomized trials [11,12]. Compared to microsurgical resection under white light, 5-ALA fluorescence-guided surgery significantly increases the rate of gross total resection, which is a strong independent predictor of survival, without increasing the rate of permanent neurological deficits [11,12]. In current standards of care, 5-ALA fluorescence-guided surgery is a relevant and cost-effective technique [12]. Fluorescein Sodium (FS), although less specific, can represent an alternative to 5-ALA for guiding the resection of HGGs. Indeed, in a multicentric phase II prospective study, FS guided-resection of HGGs resulted in a gross total resection rate of 82.6% [13]. Moreover, FS fluorescence-guided surgery increases the extent of resection, compared to microsurgical resection under white light [14,15,16]. 

In the settings of grade 2 diffuse low-grade gliomas (DLGGs), in the era of targeted therapies [17], the quality of resection still remains a critical prognosis factor. A recent and robust multicentric study including 757 *IDH*-mutant grade 2 gliomas highlighted that an extent of resection over 75% and over 80% significantly improve overall and progression-free survival, respectively. Independently of the final histo-molecular diagnosis, a post-operative tumor volume under 4.6 mL is associated with the best outcomes [18]. The introduction of brain mapping under awake conditions has undoubtedly improved the onco-functional outcomes [19]. Conversely, fluorescence-guided resection of DLGGs has raised less interest. Whereas previous reviews were focused on the surgical results provided by macroscopic fluorescence-guided resection of DLGGs [20,21,22], the aim of the present review was to summarize the limitations of macroscopic 5-ALA or FS fluorescence-guided techniques when applied to adult DLGGs and then to highlight how, in the near future, the implementation of new intraoperative technologies based on these fluorophores could challenge the limitations and improve the relevance of fluorescence for the resection of DLGGs. 

## 2. Materials and Methods

The review was not registered. A literature search was performed on the Medline electronic database, according to the PRISMA guidelines, without limitations in terms of dates. The keywords used were “glioma”, “diffuse glioma” “low-grade glioma”, “5-ALA”, “fluorescein sodium”, “fluorescence”, “fluorescence-guided resection” and “fluorescence-guided surgery”. The references of selected articles were also examined to identify additional studies including WHO grade 2 gliomas. Case reports and series dedicated mostly to HGGs but including a limited numbers of DLGGs were considered if they provided new relevant data. Animal studies, series including exclusively circumscribed gliomas (grade 1) and pediatric series were not taken into account. The PRISMA flow diagram is presented in Figure 1.

## 3. Principles of Fluorescence

### 3.1. 5-Amino-Levulinic Acid

5-ALA is the endogenous precursor of protoporphyrin IX (PpIX) which is ultimately converted into heme by ferro-chelatase in healthy cells [23]. The tumor metabolism of 5-ALA has been mainly studied in HGGs [24]. Conversely to healthy cells, glioma cells are deficient in ferro-chelatase and specifically accumulate PpIX when 5-ALA is administered in excess [24,25,26,27,28,29]. The brain–blood barrier is physiologically impermeable to 5-ALA but is disrupted in the setting of HGGs, which explains the fact that 5-ALA biodistribution after oral intake is particularly high in the cells of HGGs [23]. Moreover, membrane ABC-transporters are overexpressed in gliomas and thus facilitate 5-ALA penetration into the cells [25,30]. The negative feedback control exerted by heme on the up-stream enzymes is abolished, thereby increasing the biosynthesis of PpIX [25] (Figure 2).

As a result, in HGGs, 5-ALA works as a “chemical neuronavigation”, theoretically highlighting all tumor cells, and overcoming the limitations of MRI-based technologies whose accuracy is reduced by the intraoperative brain shift and the inability to precisely delimit the infiltration area beyond the contrast-enhanced part. Yet, PpIX concentration, and thus fluorescence intensity, depends on tumor cell density and is reduced at the periphery of HGGs [2,24,31,32]. Other parameters are likely to interfere with PpIX synthesis. For instance, tumor-independent factors such as hypoglycemia, hyperthermia and acidosis increase PpIX accumulation, while hypoxia can mildly decrease it [32,33,34,35]. 

PpIX is excitable by exposure to blue light radiation whose wavelength is adapted to its absorption spectrum (375–440 nm) [24,27,36,37]. When excited PpIX returns to its equilibrium state, it emits a red fluorescence light detectable between 620 and 704 nm (Figure 1) [28,32]. To visualize tumor fluorescence intraoperatively, the surgical microscope (Leica^®^ or Carl Zeiss^®^) has to be provided with a combination of excitation and emission filters. In the excitation light path, the proper PpIX excitation wavelength is selected by a short-pass filter. In the observer light path, a long-pass filter blocks out any excited light at wavelengths < 440 nm and only selects red porphyrin-induced fluorescence. A small fraction of excited light is also remitted from non-tumor brain parenchyma, which consequently displays a blue color in contrast to the bright-red porphyrin fluorescence of glioma [32]. During fluorescence-guided tumor resection, the neurosurgeon alternatively works under blue light to instantly visualize tumor tissue more clearly, and under white light which provides a better anatomical resolution, which is especially needed in perivascular areas.

In routine practice, 5-ALA is dissolved in tap water and orally administered at the recommended dose of 20 mg/kg bodyweight. Although mildly sour-tasting, the obtained solution is usually well tolerated [38]. The optimal level of intraoperative fluorescence is obtained when 5-ALA is administered 3 to 4 h before the induction of anesthesia. The administration of 5-ALA is contra-indicated in cases of inherited or acquired porphyria, hypersensitivity to porphyrins, severe renal or hepatic insufficiency, pregnancy and breastfeeding [4,39]. 

5-ALA administration can lead to gastro-intestinal adverse events such as nausea, vomiting and gastroesophageal reflux [40], whose frequency ranges from 0% to 3% [41]. A light-induced erythema can also be noticed in less than 1% of patients [27,41,42], and is prevented by the avoidance of intense light exposure during the 24 h following 5-ALA intake [32,38]. Biologically, the serum levels of alanine transaminase and gamma-glutamyl-transferase are elevated in 5.7% of patients after 5-ALA intake [27]. Hypoalbuminemia and hyperbilirubinemia are more rarely observed [40]. Generally, these abnormalities remain asymptomatic and spontaneously disappear within 1 to 6 weeks [43]. The postoperative levels of hepatic enzymes, bilirubin and creatinine must therefore be checked. Adverse events consecutive to 5-ALA administration are rare, minor and transient [8,11], provided that contraindications and preventive measures are respected.

### 3.2. Fluorescein Sodium

FS, whose first use for the detection of brain tumors was described in 1948 [44], is not selectively incorporated into glioma cells like 5-ALA but penetrates into the extracellular space when the blood–brain barrier is permeable [45], similarly to imaging contrast agents [46,47]. The simultaneous administration of MANNITOL increases blood–brain barrier permeability and thus FS uptake [48].

FS is excitable by exposure to wavelengths ranging from 465 to 900 nm [44,49]. When excited FS returns to its equilibrium state, it emits a yellow-green fluorescence light detectable between 520 and 530 nm [28,32]. Most surgical microscopes are provided with a “Yellow 560-module” which detects FS fluorescence.

In routine practice, FS is administered intravenously, concomitantly to anesthesia induction, at the recommended dose of 3–5 mg/kg [49,50,51,52]. At high doses (20 mg/kg), emitted fluorescence becomes visible to the naked eye [51]. Its administration is evidently contraindicated in patients who suffer from hypersensitivity to FS. 

## 4. Results Provided by Macroscopic Fluorescence-Guided Techniques Applied to DLGGs

### 4.1. Fluorescence-Guided Resection of DLGGs

#### 4.1.1. Assessment of the Rate of Positive 5-ALA Fluorescence in DLGGs 

Five studies, some also including HGGs or non-glial tumors, simply assessed the rate of 5-ALA fluorescence positivity in DLGGs. Additionally, in a study assessing the combination of 5-ALA fluorescence and intraoperative MRI for glioma resection, six DLGGs were included (Table 1). 

The reported rates of positive 5-ALA fluorescence were heterogenous, and varied widely from 0% to 100% [9,39,53,54,55,56]. Indeed, the characteristics of the patients were different from one to another study. For instance, in a study in which the fluorescence rate was of 100%, all DLGGs had the radiological features of HGGs, with a heterogeneous contrast enhancement [9]. It is important to note that the version of the WHO classification was not the same across these studies. Consequently, some gliomas classified as grade 2 would probably correspond to HGGs according to the current version of the WHO classification.

Taken together, these data indicate that only a small fraction of DLGGs could be expected to be positive for 5-ALA fluorescence. Moreover, fluorescence, if any, was only focal or vague in most cases [53,54]. For instance, among a series of 50 oligodendrogliomas, 9 (18%) tumors were fluorescent but fluorescence intensity was rarely strong (*n* = 2, 22%). Conversely, all pleomorphic xanthoastrocytomas seemed to be strongly fluorescent [39,53]. These observations justify more precisely describing the diagnostic performances of 5-ALA.
cancers-16-02698-t001_Table 1Table 1Main characteristics and results of studies assessing the rate of positive 5-ALA fluorescence in DLGGs.StudyDesignEffectivesPositive 5-ALA FluorescenceTsugu, 2011 [56]Retrospective, monocentric(WHO 2007)33 gliomas including6 (18.2%) DLGGs0/6 (0%) of DLGGs.Marbacher, 2014 [54]Retrospective, monocentric(WHO 2007)376 tumors including17 (4.5%) DLGGs8/17 (47.1%) DLGGs.Chan, 2017 [9]Retrospective, monocentric(WHO 2016)16 gliomas with heterogenous CE, finally including 3 (18.8%) DLGGsAll (100%) DLGGs.Ji, 2019 [53]Retrospective, monocentric(WHO 2016)827 presumed HGGs, finally including70 (8.5%) DLGGs11/70 (15.7%) DLGGs -9/50 (18%) oligodendrogliomas-2/20 (10%) astrocytomas, including one strong fluorescent pleomorphic xanthorastrocytoma.Goryaynov, 2019 [39]Retrospective, multicentric(WHO 2016)653 tumors, including52 (7.9%) DLGGs22/52 (42.3%) DLGGs.All pleomorphic xanthoastrocytomas (*n* = 2, 100%).Takeda, 2023 [55]Prospective, Monocentric(WHO 2021)30 deep-seated tumors, including 3 (10%) DLGGs1/3 (33.3%) DLGGs Weak fluorescence in a gemistocytic astrocytoma.CE = Contrast Enhancement.

#### 4.1.2. Assessment of the Diagnostic Performances of 5-ALA Fluorescence for the Identification of Anaplastic Foci in Diffuse Gliomas

Five prospective studies (Table 2) were designed in order to determine if 5-ALA could ease the detection of anaplastic foci in diffuse gliomas without significant contrast enhancement, and thus represent an adjunct that could help to select better tumor samples to more accurately determine glioma grade [57,58,59,60,61].

Globally, the reported sensitivity of negative 5-ALA fluorescence for the detection of DLGGs was high, and ranged from 87.5% to 100%. Conversely, positive 5-ALA fluorescence had a sensitivity ranging between 57% and 100% for the detection of HGGs [57,58,59,60,61]. This variability is inherent to the different proportions of grade 3 and grade 4 gliomas across studies, grade 3 gliomas having a lower positive fluorescence rate compared to grade 4 gliomas. Additionally, the predictive positive value of positive 5-ALA fluorescence for the detection of HGGs ranged from 85% to 100% [57,59,61]. In a recent prospective study including 89 patients with a suspected HGG, the final diagnosis consisted of 8 DLGGs and 80 HGGs (grade 3–4). 5-ALA had a sensitivity of 100% for the detection of HGGs. Interestingly, this parameter was better than that of extemporaneous examination, which was not able to distinguish between DLGG and HGG in 17.3% of cases [61]. 

In some studies, the performances of 5-ALA and metabolic imaging were compared. In a series of 38 gliomas, only 12/21 (57%) HGGs were fluorescent and contrast-enhanced, whereas 18/21 (86%) were hypermetabolic. However, 16/17 (94%) DLGGs were 5-ALA-negative and not contrast-enhanced, whereas only 10/17 (59%) were not hypermetabolic in ^18^F-FET PET [60]. In another series of 30 gliomas, areas positive for both FET-uptake and 5-ALA fluorescence were found in only 1/13 (7.7%) DLGG but in 12/17 (70.6%) HGGs. Areas positive for FET-uptake and not fluorescent were found in 6/13 (46.2%) grade 2 samples and in only 3/17 (17.6%) high-grade samples [58]. 

Taken together, these results confirm that 5-ALA has a very limited sensitivity for the detection of DLGG tissue and is less performant than metabolic imaging in this indication. Nevertheless, positive 5-ALA fluorescent has a relatively high sensibility and a high predictive positive value for the detection of grade 3 foci [58,59,60]. These results warrant a more detailed examination of the potential predictors and modulators of 5-ALA fluorescence positivity in presumed DLGGs.

#### 4.1.3. Identification of the Predictors of 5-ALA Fluorescence Positivity in DLGGs

Eleven studies (Table 3) were designed to identify the predictors of 5-ALA fluorescence positivity in gliomas [62,63,64,65,66,67,68,69,70,71,72,73]. The predictive value of some factors was also examined in previously detailed studies [57,59,60].

Regarding radiological parameters, in a series mixing 33 DLGGs and 26 HGGs, independently of the grade, 5-ALA fluorescence positivity was correlated with the intensity of contrast enhancement. Indeed, 88% of cases with focal contrast enhancement were fluorescent, whereas the rate of fluorescence dropped to 53% and 13% in cases with patchy/faint and no contrast enhancement, respectively [59]. This trend was confirmed in other studies [63,66,69,70,71,73]. In a series of 75 gliomas including 18 (24%) cases without contrast enhancement, increased tumor blood flow (assessed by arterial spin labelling) was a predictor of positive fluorescence, both in gliomas with and without contrast enhancement [67]. Focal 5-ALA-positive areas, even in the absence of gadolinium enhancement, were more frequently hypermetabolic than non-fluorescent areas [57,59,60,68,70,71].

Regarding histopathological parameters, consistently across studies, compared to 5-ALA-negative areas, 5-ALA-positive areas had an increased proliferation index [57,59,60,62,63,64,66,68,69], a more marked nuclear polymorphism [59] and a higher cell density [59,62,64]. Additionally, the microvascular density, assessed semi-quantitatively and automatically, was higher in fluorescent samples than in non-fluorescent samples [73].

The interactions between medications and 5-ALA fluorescence were explored in two studies. In a series of 27 grade 1 and 2 gliomas, 56% of patients (*n* = 15) were taking anti-epileptic drugs. Interestingly, after adjustment for confounders, positive 5-ALA fluorescence was significantly less frequently found in patients taking anti-epileptic drugs than in other patients [64]. In a bicentric retrospective study including 110 presumed DLGGs, 26 (23.6%) and 72 (65.5%) patients received preoperative dexamethasone and anti-epileptic drugs, respectively. Conversely, in multivariate analysis, corticoid and anti-epileptic drug intake were not independent predictors of tumor fluorescence [65]. The inconsistency of the results of the two last studies could be explained by the different proportions of HGG, which may constitute a confounding factor [64,65]. In accordance with the latter results, it has been established that preoperative corticoid intake does not modify the expression of genes involved in heme biosynthesis. Although corticosteroids seem to reduce PpIX efflux from tumor cells, resulting in a strengthened tumor fluorescence in glioblastoma [27,38,74], they also tighten up the blood–brain barrier [75], which should decrease 5-ALA penetration into the cells. Furthermore, preoperative anti-epileptic drug intake modifies the expression rate of two genes, having opposite effects on 5-ALA metabolism (*SLC15A1* and *ABCB6*) [76].

Some studies were dedicated to the identification of additional intrinsic and extrinsic predictors of 5-ALA fluorescence in diffuse gliomas lacking the typical radiological features of glioblastoma [63,65,66,70]. In a German series of 166 gliomas (82, 49.4% DLGGs and 84, 50.6% HGGs), age and tumor volume were predictors of intraoperative 5-ALA fluorescence, additionally to previously identified factors [70]. The proliferation index was increased in fluorescent areas, compared to non-fluorescent areas in the whole series and in grade 3, but not in grade 2, gliomas. Moreover, none of the examined molecular factors (*IDH1* mutation, 1p19q codeletion, and *MGMT* promotor methylation status) was correlated with fluorescence positivity [70], which was confirmed in another study of the same group [71]. However, in these two studies, the low proportion of IDH-wildtype gliomas compared to *IDH*-mutant may have decreased the statistical power. Indeed, in two other retrospective series which investigated more deeply the modulation of 5-ALA fluorescence positivity by molecular features, the absence of *IDH1* mutation was, conversely, an independent predictor of positive 5-ALA fluorescence [63,72]. Furthermore, the rate of *EGFR* amplification was higher in fluorescent than in non-fluorescent areas (19% vs. 5%, *p* = 0.57) [71], which is not surprising, as amplified cases would be reclassified as grade 4 according to the 2021 WHO classification [77]. It is noteworthy that, in glioblastomas, high-methylation status of the *MGMT* promotor was associated with stronger fluorescence than low-methylation status [78]. These inconsistent results regarding the link between molecular parameters and fluorescence intensity could also indicate that tumor grade is a stronger predictor of fluorescence than molecular alterations (which are not independent of the grade).

#### 4.1.4. Assessment of the Prognostic Value of Intraoperative 5-ALA Fluorescence Positivity in DLGGs

As some prognosis factors, such as the proliferation index or the presence of a hypermetabolism are predictors of fluorescence [57,59,60,62,63,64,66,68,69,70], it seems important to study the relationship between 5-ALA fluorescence and the survival outcomes. Three studies, including exclusively DLGGs, assessed whether intraoperative 5-ALA fluorescence positivity could be a predictor of survival [71,72,73] (Table 3).

In univariate analysis, 5-ALA fluorescence positivity was a predictor of shorter progression-free survival (2.3 ± 0.7 vs. 5.0 ± 0.4 years, *p* = 0.01), malignant transformation-free survival (3.9 ± 0.7 vs. 8.0 ± 0.6 years, *p* = 0.03) and overall survival (5.4 ± 1.0 vs. 10.3 ± 0.5 yeats, *p* = 0.01) [72]. The outcomes were, surprisingly, not correlated with age and histological subtypes in this survival analysis [72], raising doubts regarding the statistical robustness, notably given the restricted size of the cohort. However, the negative prognostic value of positive 5-ALA fluorescence regarding the malignant transformation-free survival [71] and overall survival [71,73] was confirmed in multivariate analysis. Well-known predictors of shorter malignant transformation-free and overall survival, such as age or *IDH1*wild-type status [71] were identified in multivariate analysis. Nevertheless, the histo-molecular diagnosis was not updated according to the 2021 WHO classification, and the *EGFR* and *TERT* status were not reported [73]. Consequently, it is highly probable that a significant number of fluorescent gliomas would now be classified as grade 3 or 4, according to the current WHO classification, leading to an overestimation of not only the rate of 5-ALA fluorescence in true DLGGs but also of its prognostic value.

#### 4.1.5. Assessment of the Usefulness of FS for the Resection of DLGGs

Four studies assessed FS fluorescence-guided resection of gliomas (Table 4) [79,80,81,82]. 

In a comparative study including a control arm, 10 gliomas were resected under FS induced fluorescence. Among these tumors, 3/4 (76%) DLGGs, which had all a contrast-enhanced area on the MRI, were fluorescent. Fluorescence intensity was correlated with cell density. No conclusion regarding the relevance of FS guided-resection of DLGGs was drawn, as the onco-functional results were not detailed by grade [79]. In a small series including five gliomas (grade 2 to 3) without contrast enhancement but with 18F FET-uptake, interestingly, positive FS was observed in each case and was either focal (corresponding to the hypermetabolic area) or diffuse. From a technical viewpoint, it was judged that FS eased the identification of anaplastic foci in the case of focal fluorescence, and of tumor borders in case of diffuse fluorescence [80]. Conversely, in a series of 28 cases, no positive fluorescence was detected in the five DLGGs, whereas all grade 3 and 4 were fluorescent. The presence of fluorescent areas was correlated with the loss of the expression of the tight-junction protein claudin-5 by endothelial cells, which can be considered a marker of blood–brain barrier disruption [81].

### 4.2. Fluorescence-Guided Biopsy of DLGGs

The technique of fluorescence-guided biopsy was developed in order to increase the diagnosis yield, notably when extemporaneous histological examination is not available [83,84,85]. A total of nine studies assessing the feasibility and the relevance of fluorescence-guided biopsy using 5-ALA fluorescence or FS included DLGGs (Table 5). For these procedures, tumor samples were examined ex vivo, in the operating room, under the appropriate wavelength, provided by a surgical microscope or a laboratory microscope [52,54,86,87,88,89,90,91,92].

#### 4.2.1. 5-ALA-Guided Biopsies

In two prospective series including tumors of various histology referred for frameless biopsy [86,88], all DLGGs, specifically sampled in hypermetabolic areas according to PET or MRI spectroscopic sequences [86], were not fluorescent, whereas all samples from grade 3 gliomas (*n* = 10) were fluorescent (strongly in 90% and weakly in 10%). In the other studies, fluorescent samples were observed in 0% to 66.6% of DLGGs and in 84.6% to 100% of grade 3–4 gliomas [54,89,90,91]. Interestingly, in case of strong fluorescence, independently of the final grade, the diagnosis yield of fluorescent samples was close to 100%. Therefore, the extemporaneous examination of strongly fluorescent samples does not seem to be mandatory, which could result not only in a significant decrease in the duration of the surgery [90,91] but also in a significant decrease in the required number of tumor samples [90]. However, the application of these conclusions to DLGGs is limited, given the low fluorescence intensity in positive cases. Moreover, for non-fluorescent cases, irrespectively of the final diagnosis, the surgery was longer than in a control group in which patients did not receive 5-ALA, because of possible resampling and trajectory adjustments [89].

#### 4.2.2. FS-Guided Biopsies

In two prospective monocentric studies, the rate of positive fluorescence ranged from 33.3% to 100% in grade 3 gliomas [52,92] and, surprisingly, reached 100% in DLGGs. However, the diagnosis of DLGG seems doubtful in 3/4 (75%) tumors, as a contrast-enhanced part was present. Preoperative corticosteroid intake seems to reduce the likelihood of obtaining a fluorescent sample, even in high-grade tumors [92]. In a retrospective study, 36% of samples from grade 2 oligodendrogliomas were fluorescent, whereas none of the samples from grade 2 astrocytomas were fluorescent. Interestingly, in this last tumor group, 48% of fluorescent samples were taken in non-contrast-enhanced areas. Conversely, some fluorescent oligodendroglioma samples were not infiltrated by the glioma [87].

According to these results, FS has a relatively good sensibility [52,92] and positive predictive value [87,92] for the detection of DLGGs. Consequently, in the case of positive fluorescence, especially if present in at least three samples from the same tumor [87], the performance of an extemporaneous histological examination does not seem to be mandatory and it is possible to reduce the number of tumor samples, leading to the same advantages as detailed before. Importantly, these different studies did not only include grade 2 and 3 gliomas, and these results are generalizable to other tumor types, notably grade 4 gliomas or lymphomas [52,87,92].

### 4.3. Main Limitations

It is difficult to assess the real fluorescence rate and relevance of macroscopic fluorescence-guided techniques in DLGGs, given the small number of studies exclusively dedicated to DLGGs, the heterogeneity of the different studies and the inconsistent assessment of fluorescence intensity. 

According to the presented studies, the rate of fluorescent DLGGs is low (135/691 = 19.5% for 5-ALA). By analogy, in malignant gliomas, whereas the tumor bulk is highly fluorescent, fluorescence intensity is weaker at the borders. The precise analysis of the latter has shown that the intensity of 5-ALA fluorescence depends on the density of tumor cells [2,6,8,41,66,93,94,95] and the proliferation index [96,97,98], which finally explains that the intensity of fluorescence depends on the tumor grade. Other factors may explain the low fluorescence rate in DLGGs. First, in DLGG samples taken in patients after 5-ALA uptake, only intravascular weak fluorescent cells were observed, whereas tumor cells were not fluorescent. Conversely, in samples from HGGs, vascular walls were discontinuous and tumor cells adjacent to vessels were strongly fluorescent [99]. These data clearly establish the link between blood–brain barrier disruption and fluorescence positivity. However, some samples taken in contrast-enhanced areas, where the blood–brain barrier is disrupted, do not emit any fluorescence [58,60,61,70], suggesting the absence of PpIX accumulation. From a metabolic viewpoint, it has been demonstrated that the level of Nicotinamide Adenine Dinucleotide (NADPH), a cofactor needed for 5-ALA metabolization, is lower in *IDH1*-mutant than in *IDH1*-wildtype cell lines, explaining the difference in fluorescence-positivity rates [100]. Additionally, the proton-dependent peptide transporter 2 (PEPT2) channel, by which 5-ALA penetrates into cells, is less expressed in DLGGs compared to HGGs [101]. Consistently, it is also less expressed in non-fluorescent grade 2 and 3 gliomas, compared to fluorescent ones [102].

Importantly, in most studies, diagnoses of DLGGs were based on previous WHO classifications. Consequently, some fluorescent DLGGs would be reclassified as grade 4 according to the current version, possibly leading to an overestimation of the rate of fluorescent-positive true DLGGs. Moreover, when present, fluorescence is generally only focal. Accordingly, the value of 5-ALA and of FS for tumor detection in DLGGs was rated at 2/5 (vs. 4/5 for 5-ALA in HGGs) by 310 neurosurgeons who participated in a survey proposed by the European Association of Neurosurgical Societies [103].

Based on the conjecture that there could be a relationship between 5-ALA plasmatic concentration and the amount of 5-ALA that would cross the blood–brain barrier, a higher dose of 5-ALA doses (40 mg/kg) was tested in 23 DLGG patients and only led to a fluorescence rate of 60% (versus 30.8% with the regular dose of 20 mg/kg) [104], with a possible decreased specificity and an increase in the rate of adverse events. 

Hence, this result confirms that macroscopic 5-ALA-fluorescence, even with optimization with the 5-ALA dose, can definitively not efficiently guide the resection of DLGGs. Therefore, the resection rates and the onco-functional outcomes reported in the literature were not commented on, all the more as most studies included tumors that were not all amenable to gross total resection, which introduce a major bias in the assessment of the extent of resection. Additionally, all these resection rates were mainly obtained from prospective studies, and there was no randomized study dedicated to the assessment of fluorescence-guided surgery in non-contrast-enhanced DLGGs. However, as 74.5% (172/231) of HGGs without contrast enhancement emit fluorescence, 5-ALA offers an opportunity to identify anaplastic foci in diffuse gliomas and could therefore improve the representativity of tumor samples in the settings of tumor biopsy or resection, but with a limited sensitivity. Compared to 5-ALA, few data regarding the use of FS for the resection or biopsy of DLGGs are reported. The rate of FS-positive fluorescence is 2-fold higher (50%, 10/20) than that of 5-ALA, but still not sufficient to guide the resection of DLGGs. These limitations explain that new intra-operative techniques have been developed to increase the sensitivity of fluorescence detection in DLGGs.

## 5. Emerging Techniques

### 5.1. Laser Spectroscopic Detection of Autofluorescence

#### 5.1.1. Proof of Concept

More than 25 years ago, the first studies carried out on gliomas, ex vivo, thanks to microspectrofluorometric techniques, and in vivo, with a spectroscopic probe, showed that normal brain and tumor tissue have distinct autofluorescence emission properties, regarding both the spectral shape and the signal amplitude, when excited at a 360 nm wavelength [105,106]. Therefore, the measure of autofluorescence by spectroscopy was combined with the acquisition of diffuse-reflectance spectra, obtained following tissue illumination by a broadband white-light source. This protocol applied ex vivo to samples of normal brain and gliomas identified a fluorescence peak around 460 nm, attributed to NAD(P)H. The intensity of diffuse reflectance between 650 and 800 nm was reduced in 20 primary and secondary brain tumors, compared to normal brain. Algorithms combining these metrics reached a respective maximal sensitivity and specificity of 97% and 96% for differentiating healthy tissue from glioma, independently of the grade [107]. Two pilot clinical studies, carried out in vivo in 39 patients, most of whom had gliomas of various grades, confirmed than a two-step empirical discrimination algorithm, based on autofluorescence and diffuse reflectance at 460 nm and 625 nm is capable of discriminating healthy brain from both tumor and infiltrated margins (confirmed by histopathological examination), with a sensitivity and a specificity of 100% and 76%, respectively [108,109]. Based on these findings, an optical spectroscopic tool measuring both white light reflectance and 337 nm excitation fluorescence spectroscopy was tested in 24 patients with glioma and 11 patients operated on for temporal lobe epilepsy. Sensitivity and specificity were 80% and 89%, respectively, for the discrimination between tumor bulk and normal parenchyma, and 94% and 93% for the discrimination between tumor infiltration and normal parenchyma [110]. Blood contamination was found to be a major confounder, justifying cleaning the probe and the cavity before measurements [108,110]. Importantly, raw fluorescence intensity was also affected by tissue optical absorption and scattering at excitation and emission wavelengths, which substantially distort the remitted signal, and photobleaching [111,112].

#### 5.1.2. Spectroscopic Signature of DLGGs

Time-resolved fluorescence spectroscopy dynamically quantifies fluorescence intensity decay in terms of lifetimes, and has an improved specificity compared to steady-state spectroscopy, as it is independent of previously cited factors that nonlinearly affect the fluorescence measurements. Consequently, time-resolved fluorescence spectroscopy was used to assess, in vivo, a series of 31 gliomas, 10 of which were grade 2. The time-resolved fluorescence emission of DLGGs excited at 337 nm was characterized by a relatively narrow broadband emission with a well-defined peak at 460 nm wavelength, corresponding to NAD(P)H. The fluorescence emission was short-lasting compared to that of the healthy brain [113]. A more precise investigation of the time-resolved fluorescence spectrum of DLGGs unveiled the fact that a 390 nm emission peak, attributable to pyridoxamine-5-phosphate or glutamate decarboxylase, was very low in DLGGs compared to normal brain or even HGGs [113,114,115]. Interestingly, an intertumoral heterogeneity was described among DLGGs, as this 390 nm peak was indeed lower and had a quicker decay in oligodendrogliomas than in astrocytomas [115]. Hence, time-resolved fluorescence spectroscopy had a sensibility and a specificity of 90% and 100%, respectively, for the distinction of DLGGs from healthy brain [113]. The development of a linear discriminant algorithm increased these values to 100% and 98%, respectively [115]. 

Multimodal two-photon endomicroscopy was used ex vivo to better characterize the signature of DLGGs. At a 275 nm excitation wavelength, Tryptophan/Collagen and Tryptophan/NADH ratios were higher in DLGGs compared to control. At a 405 nm excitation wavelength, fluorescence lifetime of porphyrins was slightly higher in DLGGs than controls or HGGs. The calculated redox ratio (flavins/(flavins + NADH)) and optical index (porphyrins/NADH) were significantly different in DLGGs compared to control. Finally, with near infra-red excitation, the optical index was also significantly different in DLGGs compared to controls. Combining the Tryptophan/Collagen ratio (275 nm), redox ratio (405 nm) and redox ratio (near infra-red), it was possible to discriminate DLGGs from both controls and HGG samples with a 100% sensitivity [116,117]. Additionally, DLGGs had a higher absorption coefficient than control tissues, although the difference was not significant [116].

These data indicate that the spectroscopic detection of autofluorescence can help to discriminate DLGGs from healthy brain.

### 5.2. Spectroscopic Detection of 5-ALA-Induced Fluorescence

#### 5.2.1. Proof of Concept

Based on the postulate that PpIX is present in DLGGs, but at a concentration too low to produce visible fluorescence, it was proposed to use spectroscopy to detect fluorescence after laser excitation at 405 nm wavelength [118,119]. Spectroscopy was used for assessment in two preliminary studies in which spectroscopic data were correlated with the histological features. First, the in vivo spectroscopic analysis of six gliomas without contrast enhancement highlighted the fact that an emission peak at 636 nm was systematically and exclusively present in regions with histologically-proven tumor cell infiltration. Conversely, the emission peak was lacking in some areas in which atypical cells, suggestive of a residual glioma infiltration, were present, resulting in a relatively low sensibility (44%) [118]. Secondly, 65 macroscopically fluorescent (*n* = 12) or non-fluorescent (*n* = 53) samples, from six grade 2-to-4 gliomas were examined ex vivo at 635 nm. Consistently with the ex vivo study, spectroscopy detected no fluorescence signal in the 36 samples that were histologically classified as infiltration-free (*n* = 28) or necrotic (*n* = 8). Fluorescence was spectroscopically detected in all of the 29 infiltrated samples. The spectroscopic fluorescence intensity was higher in macroscopically fluorescent samples than in non-fluorescent samples and higher in HGGs than in DLGGs. Consistent with macroscopic results previously detailed [57,59,60,62,63,64,66,68,69], fluorescence intensity was also independently correlated with the proliferation index [119]. Interestingly, another intensity peak was described at 585 nm and was decreased in areas of glioma infiltration, compared to adjacent healthy brain. This second peak was attributed to the presence of autofluorescent substances such as flavin, lipofuscine or NADH [120].

These findings clearly demonstrate that spectroscopy is capable of detecting glioma cells not only in tumor bulk but also in the peripheral infiltration area, independently of the grade, with a high sensibility, unveiling a powerful potential to guide the resection of DLGGs. 

#### 5.2.2. Quantification of PpIX Concentration within Glioma Tissue

Postulating that the emitted PpIX fluorescence intensity is proportional to the concentration of PpIX [97,121,122], it was proposed to quantify in vivo the absolute concentration of PpIX (C_PpIX_) in patients who received 5-ALA, using a fiberoptic probe connected to a spectrometer, with a correction for the light attenuation in tissue. 

After excitation of macroscopically non-fluorescent DLGG samples at 405 nm, visually imperceptible C_PpIX_ were detectable, whereas this was not the case in adjacent healthy brain [123,124,125]. This result confirms that the spectroscopic technique is more sensitive than macroscopic fluorescence. Interestingly, it was shown that C_PpIX_ was strongly correlated with the tissular concentration of gadolinium and with microvascular density, supporting the idea that the blood–brain barrier breakdown plays a critical role in intra-tissular accumulation of PpIX [126], consistent with macroscopic observations [69,71,73]. 

In three different studies, including, in total, 345 samples from 103 gliomas, 21 of which were DLGGs, C_PpIX_ was significantly correlated with the amount of contrast enhancement [66] and the level of visible fluorescence (strong, vague or lacking) [66,125], and was significantly higher in HGGs than in DLGGs [125]. More precisely, C_PpIX_ was independently correlated with tumor cell density, proliferation index, nuclear polymorphism and microvascular proliferation [124,125], which corroborates results obtained by non-spectroscopic studies [62,63,64,67,69,73]. This means that C_PpIX_ represents a helpful biomarker capable of predicting glioma aggressiveness. An example is provided by grade 3 gliomas, in which intra-tumoral heterogeneity was highlighted by the detection of anaplastic foci, characterized by higher C_PpIX_ compared to the rest of the tumor [124]. In terms of surgical guidance, in a series including 14 tumors, only 2 of which were DLGGs, the histopathological analysis of studied samples revealed that the spectroscopic approach had a sensibility and a specificity of 75% and 80%, respectively, for DLGG detection [123], which is obviously better than the macroscopic approach. 

While previous studies were carried out in series including gliomas of various grades, C_PpIX_ was measured in vivo in 36 samples from six grade 1 and 2 gliomas. Only 17% (4/24) of tumor-infiltrated samples were macroscopically fluorescent. Consistent with former studies, C_PpIX_ was significantly higher in fluorescent samples than in non-fluorescent samples. With a cutoff threshold of 0.0057 mg/mL, C_PpIX_ quantification had better performances than visible fluorescence, as the diagnostic accuracy reached 67%, with a sensitivity of 58% and a specificity of 83% [127]. In another series, 6/8 (75%) DLGGs had increased C_PpIX_ (>0.005 μg/mL), whereas none of these tumors were fluorescent, which provides further evidence of the increased sensitivity provided by spectroscopy compared to macroscopic fluorescence detection. Although promising, these results demonstrate that spectroscopy still imperfectly detects DLGGs, and highlight the need for more advanced protocols. 

Based on the idea that neoplastic processes provide a variety of potential biomarker targets related to hypoxia or angiogenesis, for instance, the detection of fluorescence spectra at 405 nm and of white-light reflectance spectra (450 to 720 nm) was coupled, in vivo, to quantify simultaneously C_PpIX_, PpIX photoproduct concentration, total hemoglobin concentration, oxygen saturation fraction, scattering amplitude, and scattering power, during the resection of 10 gliomas of different grades. All these parameters were significantly different between normal tissue and tumor tissue, except scattering amplitude. A biologically-relevant diagnostic algorithm combining differently expressed biomarkers was developed, and demonstrated excellent performances for the detection of DLGGs (sensibility = 94%, specificity = 100%) compared to the quantification of C_PpIX_ only (sensibility = 50%, specificity = 93%) or macroscopic fluorescence (sensibility = 8%, specificity = 100%) [128]. 

#### 5.2.3. Spectroscopic Signature of DLGGs

In aqueous solution, PpIX exists under two possible states, whose proportions depend notably on the pH [129]. As the main emission peaks of these states have different central wavelengths (620 nm and 634 nm, for alkaline and acid pH, respectively), the measure of the ratio of emitted fluorescence for both PpIX states (ratio_620/634_) was introduced (representative peaks can be found in [130]). The analysis of 35 samples from 4 gliomas showed that the ratio_620/634_ was significantly higher in DLGGs (1.62 to 1.25) and in the infiltration area of HGGs (1.04) than in the tumor bulk from HGGs (close to 0). These characteristics may be explained by a low pH in HGGs, resulting from high glycolytic activity [131]. After extraction of lipofuscin fluorescence in order to avoid an overestimation of C_PpIX_ at 620 nm, further quantifications performed both in vivo and ex vivo in ten DLGGs and HGGs confirmed these findings, with a ratio_620/634_ close to 1 in healthy tissues [132]. Consequently, the importance of this 620 nm emission peak in DLGGs suggests that the decreased fluorescence intensity observed in these tumors might not only result from a low PpIX concentration, but also from a shift in PpIX conformation, with the presence of possible aggregates explained by the microenvironmental characteristics. Pragmatically, taking into account the contributions of both states to total fluorescence could help to improve fluorescence-guided resection of DLGGs by discriminating tumor-infiltrated margins from healthy tissue [130,131,132]. The analysis of 2692 fluorescence spectra measured in vivo in 128 patients was conducted in order to unmix the two PpIX peaks and the endogenous fluorophores (NADH, lipofuscin, and flavins), and ultimately increase spectroscopic sensibility. This methodology decreased the rate of false positive-tumor identification. The ratio_620/634_, the proliferation index, and the PpIX peak blue-shift were significantly correlated with WHO grade, fluorescence visibility, and C_PpIX_ [133].

However, a recent study based both on phantoms and analysis of 200,000 spectra from 600 glioma biopsies taken in 130 patients revealed that C_PpIX_ at 620 nm does not vary much depending on pH or *IDH* status, conversely to C_PpIX_ at 634 nm, which is ultimately retained as the main factor affecting the ratio_620/634_. Moreover, doubling the 5-ALA dose significantly decreases C_PpIX_ at 620 nm, which could mean that the ratio_620/634_ depends on 5-ALA dose. Consequently, it is hypothesized that the 620 nm peak could correspond to an intermediate of heme biosynthesis, such as coproporphyrin III [134], which, however, does really not alter the relevance of considering the ratio_620/634_ for the identification of low-grade gliomas.

#### 5.2.4. Refinements and Optimization of 5-ALA-Induced Fluorescence Spectroscopic Detection

In order to improve the diagnostic accuracy of spectroscopy, it was proposed to use a machine-learning classification approach to predict tumor infiltration from the raw fluorescence. In a pilot study based on the analysis of 50 samples from 10 gliomas of various grades, three different fluorescence spectra acquired after excitation at 385 nm, 405 nm, and 420 nm were taken into account in a fully automatic clustering method. The diagnostic accuracy concerning the distinction between healthy tissue and areas of low-density tumor infiltration was improved compared to previous methods [132], and reached 77% [135].

Technically, spectroscopy only gives access to focal measurements, which are not really adapted to the surgical workflow. Two main techniques were proposed to overcome this limitation.

First, fluorescence lifetime imaging (FLIM) relies on the time delay between the excitation and fluorescence emission and offers a large field of view [136]. FLIM is conditioned by the intrinsic molecular properties and the molecular environment, but is independent of any intensity variations due to altered scattering or absorption by the tissue or the blood. Thus, FLIM was assessed ex vivo in two diffuse glioma samples. FLIM had a higher sensitivity for 5-ALA fluorescence detection than standard blue-light excitation provided by an operative microscope. However, FLIM has the disadvantage of also detecting autofluorescence. Above all, the system tested had an acquisition time of around a minute, which is not compatible with real-time surgical guidance as long as this parameter is not optimized [137]. Then, FLIM was used to simultaneously detect PpIX and NADH fluorescence in an ex vivo study, including 21 tumors, 3 of which were DLGGs. PpIX fluorescence lifetime was increased in areas of visible PpIX fluorescence, and was significantly shorter in reactive parenchyma, compared to tumor infiltrated areas. Additionally, altered NADH lifetime was typically found in reactive brain parenchyma and necrotic tumor tissue. Using a NADH/PpIX combined classifier, 93% of non-macroscopically fluorescent DLGG samples were correctly identified as pathologic [138]. The additional ex vivo analysis of 15 DLGGs, which were all non-fluorescent, by FLIM coupled with spectroscopy, highlighted the fact that tumor samples had a significantly higher PpIX fluorescence lifetime than healthy brain. However, PpIX fluorescence lifetime was variable from one DLGG to another. FLIM maps and anatomical pictures were successfully merged, allowing intraoperative guidance [139]. 

Secondly, wide-field spectrally resolved quantitative fluorescence imaging was also proposed to produce a map of PpIX across the surgical field [111,140]. According to this technique, local C_PpIX_ concentration is calculated in a pixel-wise manner across the whole wide-field image, with the application of a correction algorithm to compensate for the distortion effect of the tissue optical attenuation [140]. Thanks to processing of hyperspectral imaging data, it is also possible to decouple the contribution of PpIX from the background signal, to suppress false-positive areas caused by reflection, and to compensate for the distortive effects of blood and scattering [112]. Ex vivo, this technique is able to accurately detect non-macroscopically apparent C_PpIX_, as low as 0.01 µg/mL, which is sufficient to enhance DLGG tissue. However, this system has been tested clinically only in HGGs so far, and is limited by a long acquisition time (at least 30 s per map). Moreover, the obtained maps have only been correlated with focal spectroscopic measurements, and not with histopathological results [112,140], which will have to be carried out in the future to validate the technique.

### 5.3. Confocal Laser Endomicroscopy

Confocal Laser Endomicroscopy (CLE) technology was developed in order to provide optical biopsies, which correspond to intraoperative high-resolution pictures showing tissue architecture and cytological details at a microscopic level [141], and which could replace the extemporaneous histopathological examination. A neuropathologist should analyze the pictures, either in the operative room or using a teletransmission system, as this task can be relatively difficult for a neurosurgeon, especially in areas of low-density tumor infiltration. In some centers, CLE has already been implemented in the surgical routine. However, there should be a relatively long learning curve, both for the surgeon and the pathologist. Indeed, picture acquisition and interpretation can be challenging, because of motion artifacts or the presence of red blood cells on the probe, which can mask tumor cells.

CLE based on 5-ALA fluorescence was assessed in a small series of gliomas, including two DLGGs (Table 6). No macroscopic fluorescence was observed in these tumors. In vivo and ex vivo CLE examination of samples taken initially and at the midpoint of the resection consistently demonstrated a variable density of fluorescent cells, whereas control cortical sites were all negative for 5-ALA fluorescence. Tumor infiltration was confirmed in each case by histopathological analysis. Moreover, the histopathological and CLE examination of resection margins were in total accordance, as a residual tumor infiltration was confirmed in each sample in which cellular fluorescence persisted with a density higher than 30 cells per high-power field [142]. Conversely, ex vivo confocal laser-scanning microscopy using a band-pass filtered at 618.5–675.5 nm did not provide convincing results in DLGGs, as it detected fewer fluorescent spots than in healthy tissue [143]. However, high-resolution optical sectioning microscopy, composed of spatially separated illumination and collection beam paths, was tested ex vivo in DLGGs in a pilot study. This technique seems able to provide real-time pictures, comparable to true histopathological sections, regarding the detection of localized subcellular spots of PpIX [144]. 

More studies (*n* = 9), including non-exclusively DLGGs, assessed CLE based on FS fluorescence, performed with devices of different generations (Table 6) [142,145,146,147,148,149,150,151,152]. A feasibility study carried out on a cohort of 33 patients, including 13 DLGGs, demonstrated that these tumors did not have the same aspect as the adjacent healthy parenchyma. In areas corresponding to hyperintensity in T2-weighted MR imaging, increased cell density and modified cellular morphology were observed, in concordance with the histopathological data [153]. These patterns were identified in another series, leading to an accurate diagnosis rate of 100% for gliomas when pictures were blindly assessed by a neuropathologist [145]. Interestingly, astrocytomas and oligodendrogliomas did not have the same morphological aspect [145,153]. Four additional studies in which ex vivo CLE images were acquired after application of an external dye also reported the observation of these features [154,155,156,157]. However, it was not possible to precisely estimate the grade of some gliomas based on CLE pictures, as mitosis, endothelial proliferation and necrosis were sometimes not clearly assessable, although cell density was higher in confirmed DLGGs than in HGGs [146,147].

The diagnostic performances of in vivo CLE imaging and of frozen sections for the detection of DLGGs and HGGs were compared in a recently published study. For the detection of DLGGs, CLE had a good specificity (99%) but its sensitivity (56%) was lower than that of frozen sections (78%). For the detection of HGGs, frozen sections still had a better sensitivity than CLE [152]. In another series of 32 gliomas, including 3 DLGGs, ex vivo CLE had a sensitivity and a specificity of 66% and 94%, respectively [148]. Additionally, in a series of 74 tumors, including 8 DLGGs, CLE detected tumor in all cases. According to the analysis of a mean number of 372 in vivo and ex vivo pictures per patient, the CLE sensitivity and specificity for glioma detection reached 91% and 94%, respectively, which was in line with the performances of frozen sections, as the corresponding values were 97% and 93%, respectively [146]. Interestingly, in a series of gliomas, including only two DLGGs, the review of CLE pictures from resection margins by four neuropathologists concluded that CLE and histological sections had a concordance rate of 61.6%. CLE had a relatively good sensitivity for the detection of residual tumor infiltration (79%) but a very low specificity (37%) [151].

These variabilities in terms of performances could be explained by a difference in training for CLE picture analysis, but also in the mode of picture acquisition. Indeed, the analysis of 17,951 and 10,125 pictures, respectively taken ex vivo and in vivo, highlighted the fact that in vivo pictures had significantly higher brightness and contrast values than ex vivo pictures, and thus better diagnostic performances for glioma detection [150]. The quality of ex vivo pictures depended on the time elapsed since FS injection [150], while this effect was less marked for in vivo pictures [149,150]. Interestingly, an intraoperative re-injection of FS could decrease the rate of non-diagnostic pictures from 50%, particularly in the case of ex vivo acquisition [148,150,158]. These data support the fact that in vivo acquisition has to be preferred to ex vivo acquisition whenever possible.

In trained surgical teams, CLE use was not associated with a significant increase in surgery duration [146,149,152,153] and, finally, provided quicker results than the examination of frozen sections [152].

The real usefulness of CLE for the resection of DLGGs, whatever the dye used, is not clearly proven yet, given the very restricted number of cases included in previous studies. Nevertheless, these different data are promising and suggest that CLE could be helpful intraoperatively during the management of DLGGs (1) to ease the choice of biopsy sites, (2) to provide extemporaneous histological diagnosis and (3) to analyze the resection margins in real time, which could impact the surgical strategy.
cancers-16-02698-t006_Table 6Table 6Characteristics of studies assessing the performances of 5-ALA and FS for the resection of DLGGs.StudyDesignEffectivesMain Conclusions5-ALASanai, 2011[142]Prospective,monocentricin vivo + ex vivo10 gliomas, including 2 (20%) DLGGsNo macroscopic fluorescence.100% in vivo and ex vivo fluorescence for superficial and tumor core samples.Perfect concordance with histopathological analysis for margin samples.FLUORESCEIN SODIUMSanai, 2011[153]Prospective,monocentricin vivo(WHO 2007)33 tumors, including 13 (39.4%) DLGGsFeasibility study.Increase in surgical duration of 15–20 min.Correct identification of tumor margin.Morphological aspect consistent with histopathological sections.Eschbacher, 2012 [145]Prospective,monocentricIn vivo(WHO 2007)50 tumors, including 8 (16%) DLGGsCell density and atypia well correlated with histological sections.Astrocytoma cells more elongated and atypical than oligodendroglioma cells.Blinded analysis: 4/4 (100%) accurate diagnosis for gliomas.Martirosyan, 2016 [146]Prospective,monocentricin vivo + ex vivo(WHO 2007)74 tumors, including 21 gliomas(8 DLGGs)Mean duration of 5.8 min per patient.Performances for detection of gliomas (all grade):Sensitivity = 91%, Specificity = 94%.Precise estimation of the grade not possible in all cases.Pavlov, 2016[147]Prospective,monocentricin vivo(WHO 2007)9 tumors, including 2 (22.2%) DLGGsResection or biopsyTumor detected in all cases but impossible to precisely estimate the grade, as expected criteria were not clearly identified (mitosis, endothelial proliferation, andnecrosis).Belykh, 2020[148]Prospective,monocentricEx vivo(WHO 2016)47 tumors, including 32 gliomas (3 DLGGs)Performances for detection of gliomas, independently of the grade: Sensitivity = 66%, Specificity = 94%.Fluorescein Sodium re-injection: more pictures with accurate diagnosis (67% to 93%) and fewer non-diagnostic pictures (26% to 13%)Höhne, 2021[149]Retrospective,monocentricin vivo(WHO 2016)12 tumors, including 1 (8.3%) grade 2 oligodendrogliomaMacroscopic fluorescence visible at the tumor center and borders but not in the perilesional zone.Confirmation of abnormal aspects in these areas compared to adjacent brain.Timing of dye injection: no impact on picture quality.Xu, 2022[150]Re-analysis of 2 monocentric prospective seriesin vivo + ex vivo(WHO 2016)73 tumors, including 42 gliomasCompared to ex vivo pictures, in vivo pictures have significantly higher brightness and contrast values and better diagnostic performances.For ex vivo pictures: negative correlation between contrast and time from dye injection.Xu, 2024[151]Retrospective,bicentricin vivo(WHO 2021)28 gliomas, including 2 (7.1%) DLGGsReview of CLE pictures from resection marginsConcordance of CLE and histological sections: 61.6%.CLE: Sensitivity = 79%, Specificity = 37%PPV = 65%, PNV = 53%.Wagner, 2024[152]Prospective,tricentricin vivo(WHO 2021)203 tumors, including 9 (4.4%) DLGGs and 77 (37.9%) HGGsDLGGsSensitivity: 56% (CLE) vs. 78% (frozen sections).Specificity: 99% (CLE) vs. 99% (frozen sections).HGGsSensitivity: 86% (CLE) vs. 94% (frozen sections).Specificity: 95% (CLE) vs. 100% (frozen sections).Median assessment3 min (CLE) vs. 27 min (frozen sections).

## 6. Conclusions

Macroscopic data clearly indicate that 5-ALA fluorescence is helpful for identifying anaplastic foci within diffuse gliomas with a relatively good sensitivity, but is not suitable to guide DLGG resection or biopsy (Figure 3). Importantly, the presence of intra-tumoral 5-ALA fluorescence within a non-contrast-enhanced glioma is a negative prognostic factor. Spectroscopic detection of 5-ALA-induced fluorescence is capable of detecting very low and non-macroscopically visible concentrations of protoporphyrin IX and has excellent performances for the detection of DLGGs. Moreover, DLGGs have a specific spectroscopic signature, characterized by two fluorescence emission peaks, which is useful for distinguishing them not only from healthy brain but also from HGGs.

Few data regarding the use of FS in DLGGs are available, but they indicate that this fluorophore is also not suitable to relevantly guide DLGG resection or biopsy. However, FS is the preferred dye for CLE (Figure 3). This technique is promising, but its application to DLGGs has not really provided equivalent results to the extemporaneous histological examination so far, and its implementation in the clinical routine is associated with a relatively long learning-curve. Indeed, only teams accustomed to analyzing CLE pictures for a long time managed to obtained good diagnostic performances.

## 7. Future Directions

Intraoperative spectroscopic approaches undoubtedly provide very promising results for the intraoperative identification of DLGGs, with a true potential to guide tumor resection. First, techniques such as hyperspectral imaging could be assessed for simultaneously measuring autofluorescence and induced-fluorescence using different excitation wavelengths, in order to refine the characterization of the DLGG optical signature. By providing quantitative information about the chemical composition with high-spectral resolution, such technologies would help to more precisely study the intertumoral heterogeneity regarding fluorescence signature, for instance, by comparing oligodendrogliomas and astrocytomas spectra and the intratumoral heterogeneity. Second, as spectroscopic probes only give access to very localized analysis, it is required to continue the development of wide-field spectroscopic systems, to fit with the intraoperative workflow in the setting of DLGG resection. Hyperspectral imaging could also give the possibility of reconstructing 2D optical images. These two goals motivated the initiation of the European HYPERPROBE study, with the ultimate objective of developing a new intraoperative tool [159,160]. Additionally, new hyperspectral cameras, based on the single-pixel technique, could be used to increase the quality of the pictures. Conversely, spectroscopic probes are adapted for fluorescence-guided biopsy [161,162].

Regarding CLE, prospective studies are needed to evaluate the diagnostic performances for the detection of tumor infiltration at the margins of the resection cavity. Moreover, new systems based on the detection of tumor autofluorescence could also be interesting, giving the opportunity to avoid the administration of a dye, and deserve further investigation [163,164]. Moreover, the future implementation of deep-learning technologies could improve the diagnostic precision of CLE [165].

Finally, the use of new fluorophores could also be considered. Indocyanin green (ICG) was initially assessed in 1996 and differences in the dynamic optical signals among normal brain and histologically-proven DLGGs were highlighted at the cortical and sub-cortical levels [166]. More recent studies were based on the second-Window-ICG (SWIG) technique, which takes advantage of the increased endothelial permeability in peritumoral tissue, resulting in a delayed intracellular accumulation of ICG in these areas for intraoperative visualization of the tumor [167,168,169,170]. SWIG can localize contrast-enhanced gliomas, but all the grade 3 and 2 gliomas (*n* = 3) were not visible [171]. Of note, the use of ICG angiography can be proposed to monitor the blood flow in peritumoral vessels during the resection of tumors, including DLGGs [172] The obtained results are not detailed, as this technology is not really within the scope of this review. Tozuleristide (BLZ-100) is another near-infrared imaging-agent candidate for fluorescence-guided surgery. A phase 1 study including 17 gliomas has been carried out in order to determine the best posology. However, for the highest doses, intraoperative and ex vivo fluorescence was present in only 42.9% of DLGGs [173].

## Figures and Tables

**Figure 1 cancers-16-02698-f001:**
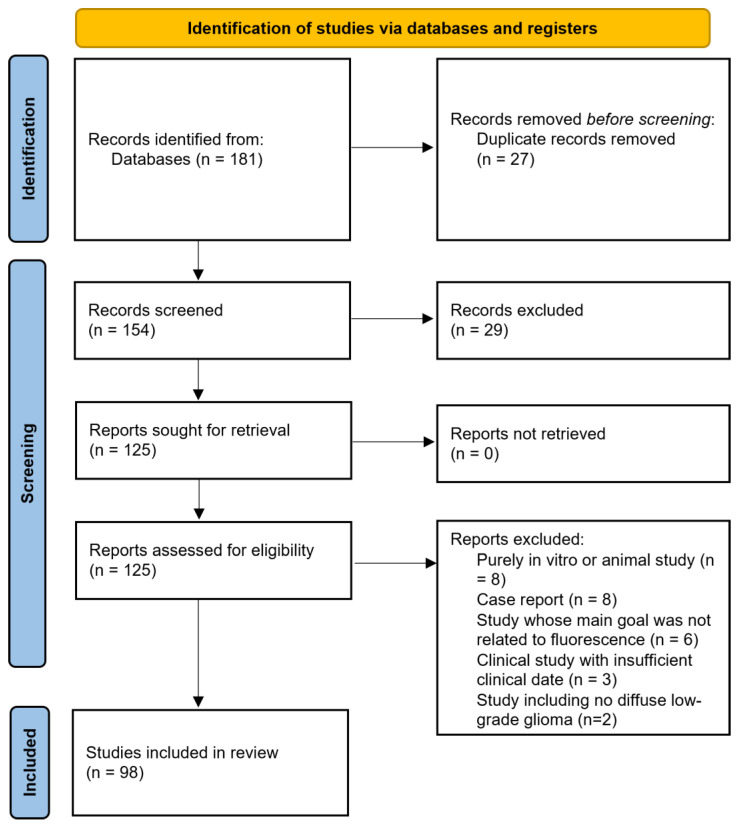
Standard PRISMA flow diagram.

**Figure 2 cancers-16-02698-f002:**
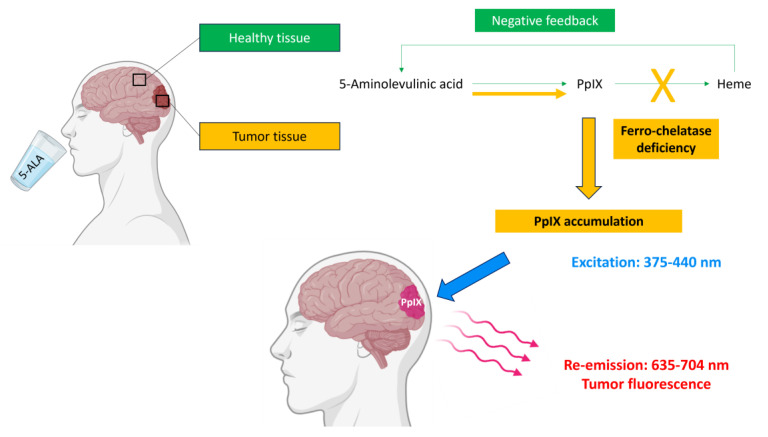
Mechanisms leading to tumor detection using 5-ALA (partly created with Biorender).

**Figure 3 cancers-16-02698-f003:**
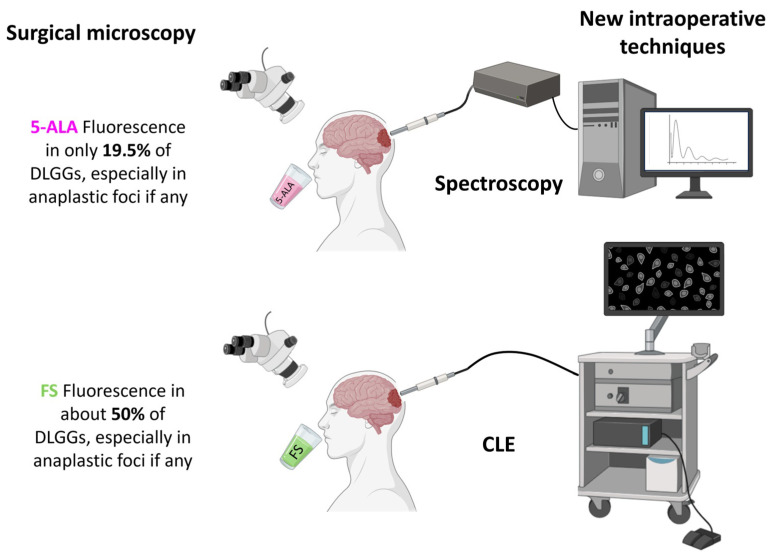
Summary of results provided by macroscopic fluorescence-guided surgery in DLGGs and emerging new intraoperative techniques able to challenge the current limitations (partly created with Biorender).

**Table 2 cancers-16-02698-t002:** Main characteristics and results of studies assessing the diagnostic performances of 5-ALA for the identification of anaplastic foci in DLGGs.

Study	Design	Effectives	Main Conclusions
Widhalm, 2010 [57]	Prospective, monocentric ^11^C Methionine PET (WHO 2007)	17 gliomas without significant CE: 8 (47.1%) grade 2 9 (52.9%) grade 3	Performance of negative fluorescence for low-grade detection Sensibility = 100%. PPV = 89%. Performance of fluorescence for high-grade detection Sensibility = 89%. PPV = 100%.
Floeth, 2010 [60]	Prospective, multicentric ^18^F FET PET (WHO 2007)	38 gliomas with CE in 13 (43.3%) cases: 17 (44.7%) grade 2 19 (50%) grade 3 2 (5.3%) grade 4	Performance of negative fluorescence for low-grade detection Sensibility = 94%. Performance of fluorescence for high-grade detection Sensibility = 57%. Hypermetabolism in ^18^F-FET 7/17 (41%) grade 2 gliomas. 18/21 (86%) grade 3–4 gliomas.
Ewelt, 2011 [58]	Prospective, monocentric ^18^F FET PET (WHO 2007)	30 gliomas with CE in 12 (40%) cases: 13 (43.3%) grade 2 15 (50%) grade 3 2 (6.7%) grade 4	Performance of negative fluorescence for low-grade detection Sensibility = 92.3%. Performance of fluorescence for high-grade detection Sensitivity = 70.6%. Specificity = 92.3%. Negative fluorescence and positive FET uptake 6/13 (46.2%) grade 2 gliomas. Positive fluorescence and FET uptake 1/13 (7.7%) grade 2 gliomas.
Widhalm, 2013 [59]	Prospective, monocentric ^18^F FET or ^11^C Methionine PET (WHO 2007)	59 gliomas without significant CE 33 (55.9%) grade 2 26 (44.1%) grade 3	Performance of negative fluorescence for low-grade detection Sensibility = 91%. Performance of fluorescence for high-grade detection Sensitivity = 88%. Specificity = 89%. PPV = 85%.
Watts, 2023 [61]	Prospective, tricentric (WHO 2021)	89 suspected HGGs, including finally 1 (1.1%) grade 1 8 (9.0%) grade 2 3 (3.4%) grade 3 77 (86.5%) grade 4	Performance of negative fluorescence for low-grade detection Sensibility = 87.5%. Performance of fluorescence for high-grade detection Sensitivity = 100%. Specificity = 88.9%. PPV = 98.8%. NPV = 100%.

CE = Contrast Enhancement, FET = Fluoro-Ethyl-Tyrosine, NPV = Negative Predictive Value, PET = Positron Emission Topography, PPV = Positive Predictive Value.

**Table 3 cancers-16-02698-t003:** Main characteristics and results of studies assessing the predictors and modulators of fluorescence positivity in TDLGGs.

Study	Design	Effectives	Predictors of Intraoperative Fluorescence
Arita, 2012 [62]	Prospective, monocentric ^11^C Methionine PET (WHO 2007)	11 gliomas, including 2 (18.2%) grade 2 (peripheric samples)	Fluorescence and ^11^C-methionine uptake are independently associated with cell density. Fluorescence is associated with proliferation index and cell density.
Jaber, 2016 [70]	Retrospective analysis of a prospectively collected database, monocentric (WHO 2007)	166 gliomas lacking the typical presentation of glioblastoma 82 (49.4%) grade 2 76 (45.8%) grade 3 8 (4.8%) grade 4	Age, tumor grade, tumor volume, contrast enhancement and ^18^F-FET uptake.
Saito, 2017 [63]	Retrospective, Monocentric (WHO 2007)	60 gliomas 8 (13.3%) grade 2 17 (28.3%) grade 3 35 (58.3%) grade 4	In univariate analysis: *IDH1*wt, no 1p19 codeletion, proliferation index, tumor margin heterogeneity, contrast enhancement. In multivariate analysis: *IDH1*wt.
Jaber, 2018 [71]	Retrospective analysis of a prospectively collected database, monocentric (WHO 2016)	74 DLGGs 12 (16.2%) oligodendrogliomas 62 (13.8%) astrocytomas	In univariate and multivariate analysis: FET uptake and preoperative contrast enhancement.
Goryaynov, 2019bis [64]	Retrospective, Monocentric (WHO 2016)	27 gliomas including 22 (81.5%) grade 2	Cell density, proliferation index, anti-epileptic drug intake.
Widhalm, 2019 [66]	Prospective, monocentric (WHO 2016)	22 suspected DLGGs 8 (36%) grade 2 11 (50%) grade 3 3 (14%) grade 4	Contrast enhancement, increased cell density.
Wadiura, 2020 [65]	Retrospective, Bicentric (WHO 2016)	110 suspected DLGGs 65 (59%) grade 2 38 (35%) grade 3 7 (6%) grade 4	Dexamethasone/anti-epileptic drugs intake were not independent predictors.
Batalov, 2021 [67]	Retrospective, Monocentric (WHO 2016)	75 gliomas with CE in 57 (76%) cases: 16 (21.3%) grade 2 13 (17.3%) grade 3 46 (61.4%) grade 4	Increased Tumor Blood Flow (assessed by Arterial Spin Labelling) is a predictor of positive fluorescence, both in gliomas with and without contrast enhancement.
Kaneko, 2021 [68]	Retrospective, Monocentric (WHO 2016)	25 DLGGs initially suspected to be high-grade (24% CE) 8 (32%) oligodendrogliomas 15 (60%) *IDH*mut and 2 (8%) *IDH*wt astrocytomas	In univariate analysis: gadolinium enhancement, proliferation index, 18F-FET PET uptake ratio and ADC-based tumor cellularity In multivariate analysis: proliferation index and 18F FET PET uptake ratio.
Hosmann, 2021 [72]	Retrospective, bicentric (WHO 2016)	59 DLGGs 29 (49%) *IDH1*mut and 3 (5%) *IDH1*wt astrocytomas 23 (39%) Oligodendrogliomas 4 (7%) Not otherwise specified	*IDH1*wt status significantly more frequent in fluorescent tumors than in non-fluorescent tumors.
Müther, 2022 [69]	Retrospective, Monocentric (WHO 2016)	179 gliomas 113 (63.1%) grade 2 66 (36.9%) grade 3	Contrast enhancement on the MRI, proliferation index.
Hosmann, 2023 [73]	Retrospective, 3 centers (WHO 2016)	86 DLGGs with CE for 23 (26.7%) cases 56 (65.1%) astocytomas 30 (34.9%) oligodendroliomas	Contrast enhancement and CD34 expression correlated with fluorescence positivity.
**TOTAL**	** Total: ** Table 1, Table 2 and Table 3		** Positive 5-ALA fluorescence **
	**659 DLGGs**		**19.4% (128/659) DLGGs**
	**231 grade 3 (all)**		**74.5% (172/231) grade 3 (all) and**
	**and grade 4 (without CE) gliomas**		**Grade 4 (without CE) gliomas**

CE = Contrast Enhancement, FET = Fluoro-Ethyl-Tyrosine, *IDH*mut = *IDH*1 mutated, *IDH1*wt = *IDH1* wild-type, PET = Positron Emission Topography.

**Table 4 cancers-16-02698-t004:** Characteristics of studies assessing the role of FS for the resection of DLGGs.

Study	Design	Effectives	Main Conclusions
STUDIES ASSESSING RESECTION GUIDED BY FLUORESCEIN SODIUM			
Chen, 2012 [79]	Prospective, monocentric Dose: 15–20 mg/kg (WHO 2007)	10 gliomas, including 4 (40%) grade 2	FS fluorescence positivity 3/4 (75%) grade 2. Contrast enhancement in all fluorescent DLGGs but not in the remaining one.
Schebesch, 2013 [82]	Retrospective, monocentric Dose: 3–4 mg/kg (WHO 2007)	26 gliomas, including 3 (11.5%) grade 2	FS fluorescence Helpful in 2/3 DLGGs.
Schebesch, 2018 [80]	Retrospective, monocentric, Dose: 5 mg/kg (WHO 2016)	5 gliomas without contrast enhancement but with ^18^F FET uptake, including 1 (20%) grade 2 3 (60%) grade 3	FS fluorescence positivity 100% cases (diffuse or focal).
Xiang, 2018 [81]	Retrospective, monocentric Dose: 5 mg/kg (WHO 2016)	28 gliomas 5 (17.9%) grade 2 6 (21.4%) grade 3 17 (60.7%) grade 4	FS fluorescence positivity 0/5 (0%) grade 2. Significant decrease in Claudin-5 expression by endothelial cells in fluorescent gliomas.
**TOTAL**			** Positive FS fluorescence **
	**13 DLGGs**		**46% (6/13) DLGGs**
	**5 grade 3 gliomas without CE**		**60% (3/5) grade 3 gliomas without CE**

CE = Contrast Enhancement.

**Table 5 cancers-16-02698-t005:** Characteristics of studies assessing the performances of 5-ALA for the biopsy of DLGGs.

Study	Design	Effectives	Ex Vivo Positive 5-ALA Fluorescence at Target
5-ALA-GUIDED BIOPSIES			
Von Campe, 2012 [88]	Prospective, monocentric Frameless biopsies (WHO 2007)	14 tumors, including 2 (14.3%) DLGGs	In 0/2 (0%) DLGGs.
Widhalm, 2012 [86]	Prospective, monocentric Frameless biopsies (WHO 2007)	50 tumors, including 6 (12%) DLGGs	In 0/19 (0%) samples from DLGGs.
Marbacher, 2014 [54]	Retrospective, monocentric Frameless biopsies (WHO 2007)	82 tumors, including 12 (14.6%) DLGGs	In 3/12 (25%) DLGGs.
Shofty, 2019 [89]	Retrospective, monocentric Frameless biopsies (WHO 2007)	34 tumors, including 3 (8.8%) DLGGs	In 2/3 (66%) DLGGs.
Millesi, 2020 [90]	Prospective, monocentric Frameless biopsies (WHO 2007 & 2016)	79 tumors, including 6 (7.6%) DLGGs	In 2/6 (33.3%) DLGGs (vague).
Malinova, 2020 [91]	Retrospective, monocentric Frame-based biopsies (WHO 2016)	39 tumors, including 3 (7.7%) DLGGs	In 0/3 (0%) DLGGs.
**TOTAL**		**32 DLGGs**	** Positive 5-ALA fluorescence ** **21.9% (7/32) DLGGs**
FS-GUIDED BIOPSIES			
Thien, 2018 [92]	Prospective, monocentric Frameless biopsies Dose: 5 mg/kg (WHO 2016)	18 tumors with CE, including 3 (16.7%) DLGGs	In 3/3 (100%) samples from DLGGs.
Nevzati, 2020 [52]	Prospective, monocentric Frameless biopsies Dose: 3 mg/kg (WHO 2016)	17 tumors, including 1 (5.9%) DLGG	In all (100%) DLGGs.
Xu, 2022 [87]	Retrospective, monocentric Frameless and frame-based biopsies Dose: 5 mg/kg (WHO 2016)	44 tumors, including 3 (6.8%) DLGGs (one astrocytoma, 2 oligodendrogliomas)	0% of samples from grade 2 astrocytoma 36% of samples from grade 2 oligodendrogliomas, but 9% of samples were not glioma-infiltrated
**TOTAL ***		**7 DLGGs**	** Positive FS fluorescence ** **57.1% (4/7) DLGGs**

* Excluding cases from [87], as the number of fluorescent samples was indicated but not that of fluorescent tumors.

## Data Availability

No new data were created or analyzed in this study. Data sharing is not applicable to this article.

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
