# Peer review of "Fluorescence-Guided Surgical Techniques in Adult Diffuse Low-Grade Gliomas: State-of-the-Art and Emerging Techniques: A Systematic Review"

_cancers, 2024, doi:10.3390/cancers16152698_

Round 1

Reviewer 1 Report

Comments and Suggestions for Authors

1.      The overall topic has been reviewed before and even very recently in Cancers (Bianconi et al., 2023).

2.      I feel that this is actually (almost) a systematic review of the use of 5-ALA and fluorescein sodium in LGG surgery, and I would encourage the authors to say so and provide the standard PRISMA diagram. Even if the authors have really missed a few publications, it will be easy to incorporate the respective information. This will result in a much more authoritative review.

3.      It might be useful to include few comments on other fluorophores such as ICG, even if there is only scarce (and negative) data.

As pointed out above, the use of 5-ALA and fluorescein sodium in LGG surgery has been reviewed systematically before and therefore this part of the paper does not really add significantly to the literature. However, the present paper is very comprehensive and the Emerging techniques section deals with very timely issues. The paper is well written and illustrated.

Author Response

  1. The overall topic has been reviewed before and even very recently in Cancers (Bianconi et al., 2023).

We thank reviewer 1 for this first comment. We agree that several reviews regarding the use of fluorescence for the surgery of diffuse gliomas have already been published. The review from Bianconi et al was indeed cited in our manuscript (reference 22). Whereas Bianconi et al’s review was mainly centered on surgical results provided by macroscopic fluorescence techniques, the originality of the present review was to summarize the limitations of these techniques when applied to adult low-grade gliomas and then to highlight how implementation of technological refinements could improve the relevance of the technique in this indication, in the near future. The introduction was modified to better highlight this specificity.

  1. I feel that this is actually (almost) a systematic review of the use of 5-ALA and fluorescein sodium in LGG surgery, and I would encourage the authors to say so and provide the standard PRISMA diagram. Even if the authors have really missed a few publications, it will be easy to incorporate the respective information. This will result in a much more authoritative review.

This review was indeed prepared as a systematic review with a very rigorous literature analysis but we initially preferred to designate it as a narrative review because different questions were addressed (limitation of macroscopic techniques / new intraoperative techniques). Following reviewer 1 suggestion, the title was modified to present this paper as a systematic review. The method section has been corrected accordingly and a standard PRISMA diagram was added as well (Figure 1).

  1. It might be useful to include few comments on other fluorophores such as ICG, even if there is only scarce (and negative) data.

We thank reviewer 1 for raising this point. Comments on other fluorophores were present in the initial manuscript, at the end of the section regarding future directions (7). Regarding ICG, all available data are presented. There is an additional article from Ferroli et al (2011, acta neurochirurgica suppl) but the goal of this article was to assess the relevance of ICG angiography during tumor resection for monitoring the blood flow in peritumoral vessels. According to us, this is not under the scope of the study. However, this reference is not cited in the above-mentioned section.

Reviewer 2 Report

Comments and Suggestions for Authors

The authors provided an overview of fluorescence-guided surgical techniques for diffuse low-grade gliomas, highlighting the limitations of existing methods and discussing emerging technologies that may offer new opportunities for improved surgical outcomes. However, some minor revisions might improve the literature review, the paper could make an even stronger contribution to the field.

1.  Lack of comparison between fluorescent surgery and non-fluorescent surgery, as well as the assessment of patient tumor resection rates and short-term/long-term prognosis relationships.

2.  While the authors mention the limitations of existing fluorescence-guided techniques, they could provide more specific examples, figure or data to illustrate these challenges. This would help readers understand the complexities of using these techniques for low-grade gliomas and appreciate the need for new approaches.

3.  Limitations were mentioned by the authors, however more potential methods could be enumerated to improve the fluorescence technique of low-grade gliomas, thereby enhancing the application value of future fluorescence imaging technologies in the treatment of LGG tumors.

Author Response

The authors provided an overview of fluorescence-guided surgical techniques for diffuse low-grade gliomas, highlighting the limitations of existing methods and discussing emerging technologies that may offer new opportunities for improved surgical outcomes. However, some minor revisions might improve the literature review, the paper could make an even stronger contribution to the field.

We thank Reviewer 2 for this positive feedback.

  1. Lack of comparison between fluorescent surgery and non-fluorescent surgery, as well as the assessment of patient tumor resection rates and short-term/long-term prognosis relationships.

We thank reviewer 2 for raising this important point that was already addressed in the first version of the manuscript (line 415-422). We voluntarily chose not to compare the results provided by conventional surgery and fluorescence-guided surgery in terms of resection rates and outcomes for different reasons. First of all, the review from Bianconi et al (mentioned by reviewer 1) has already made this comparison. Most importantly, it clearly appears that macroscopic fluorescence is not able to guide the resection of DLGGs. Then, all resection rates were obtained in non-randomized studies. Above all, studies comparing the two techniques included gliomas located in eloquent areas, which were non-consistently amenable to total resection. This last point introduced a major bias in the evaluation of the surgical results. These different points have been detailed in the manuscript.

  1. While the authors mention the limitations of existing fluorescence-guided techniques, they could provide more specific examples, figure or data to illustrate these challenges. This would help readers understand the complexities of using these techniques for low-grade gliomas and appreciate the need for new approaches.

We agree that it is relevant to enhance the message regarding the limitations of fluorescence-guided techniques in DLGGs. As an illustrative case seems few useful (surgical view with no fluorescence), we sought to propose a summary of the limitations and of the new techniques capable of challenging these limitations in a new figure (Figure 3) which is now included in the manuscript.

  1. Limitations were mentioned by the authors, however more potential methods could be enumerated to improve the fluorescence technique of low-grade gliomas, thereby enhancing the application value of future fluorescence imaging technologies in the treatment of LGG tumors.

According to our knowledge, all methods and strategies (including increase in fluorophore doses) capable of improving the fluorescence techniques, have been reviewed and discussed. However, it is true that other methods such as Raman microspectroscopic imaging have not been reviewed as they did not fall under the scope of this study.

Round 2

Reviewer 1 Report

Comments and Suggestions for Authors

 I have no further concerns.